# Geospatial variation in dietary patterns and their association with heart disease in Bangladeshi population: Evidence from a nationwide survey

**Rafid Hassan[1,2], Masum Ali[3], Sanjib Saha[4], Sadika Akhter[2,5], Md. Ruhul Amin[1]\***

**1** Institute of Nutrition and Food Sciences, University of Dhaka, Dhaka, Bangladesh, **2** International Centre for Diarrhoeal Disease Research, Bangladesh (icddr,b), Dhaka, Bangladesh, **3** International Food Policy Research Institute (IFPRI), Dhaka, Bangladesh, **4** Department of Clinical Science (Malmö), Health Economics Unit, Lund University, Lund, Sweden, **5** Faculty of Health, School of Health and Social Development, Deakin University, Burwood, Australia

* ruhul.infs@du.ac.bd

**Data Availability Statement:** This research was carried out using the 2016-2017 Bangladesh Household Income and Expenditure Survey (HIES)

## Abstract

Heart disease is a significant public health threat, and its burden is increasing worldwide. Recent evidence suggests that dietary pattern is a key modifiable factor for heart disease. Research regarding dietary patterns and heart disease in Bangladesh with their spatial variability is limited. In this study, the spatial variation and relationship between dietary patterns and heart disease among Bangladeshi people was investigated. The country-representative Household Income and Expenditure Survey 2016 dataset was used, and a total of 77,207 participants aged 30 years and over were included. A principal component analysis was conducted to derive the dietary patterns. Both statistical and spatial analyses were performed. The overall prevalence of heart disease was 3.6%, with a variation of 0.6% to 10.4% across districts of Bangladesh. Three major dietary patterns, named "festival pattern", "pickles and fast foods pattern", and "rice and vegetable pattern" were identified, accounting for 25.2% of the total dietary variance. Both the dietary pattern and heart disease rate varied across the region. A higher risk of heart disease was persistent in the western-south, southern, central, and eastern regions, as was greater adherence to the "festival pattern" and "pickles and fast foods pattern." After adjusting for confounders, participants with the highest adherence to the "rice and vegetable pattern" were associated with a lower likelihood of developing heart disease (AOR: 0.78, 95% CI: 0.64–0.95, p <0.05), while the highest adherence to the "pickles and fast foods pattern" was associated with a higher likelihood of developing heart disease (AOR: 1.50, 95% CI: 1.27–1.76, p <0.001). The spatial disparities in the prevalence of heart disease and dietary patterns underscore the significance of prioritizing intervention at the district level, especially in the western-south, southern, central, and eastern regions, to control the rising heart disease trends in Bangladesh.

data which was conducted by the Bangladesh Bureau of Statistics (BBS) with technical and financial support from the World Bank. However, the BBS has imposed legal restrictions to prevent sharing the data publicly. Data is available upon request to the corresponding author with the permission from the BBS (Director General, Bangladesh Bureau of Statistics, dg@bbs.gov.bd, +88-02-5500-7056, www.bbs.gov.bd).

**Funding:** The study was funded by the National Science and Technology (NST) Fellowships 2020-2021, under Ministry of Science and Technology, Bangladesh. The funders had no role in study design, data collection and analysis, decision to publish, or preparation of the manuscript.

**Competing interests:** The authors have declared that no competing interests exist.

## Introduction

Heart disease is escalating globally, a number one cause of death and disabilities that exacerbate health and well-being [1]. Over the past decades, the rate of heart disease has increased by 103%, affecting 197 million people and contributing to 182 million disability-adjusted life years and 9.1 million deaths (16% of total deaths) in 2019 [1,2]. The low and middle-income countries (LMICs) bear the greatest burden from the boom in heart disease rate due to poor medical care and other lifestyle-related factors, with over 80% of all heart disease-related deaths and disabilities [3]. South Asian countries have the highest burden and mortality from heart disease among all LMICs (40–60% higher risk of mortality), accounting for more than one-fourth of cardiovascular diseases (CVDs)-related deaths [4,5]. Furthermore, South Asian ethnicities are an independent risk factor for heart disease, with a 3 to 5 times higher chance of developing heart disease, and heart disease manifest 5 to 10 years earlier than in Western countries. However, the rate of incidence of CVDs was highest among Bangladeshis and exposed to heart disease at an earlier age [6,7]. Depending on context, the prevalence of heart disease rates in Bangladesh varies significantly, ranging from 1.85 to 78% [8,9]. According to a systematic review and meta-analysis, the weighted pool prevalence of coronary heart disease in Bangladesh was 4%, ischemic heart disease was 2%, and heart attack was 2% [10].

The dietary components significantly influence the prognosis of heart disease. A healthy diet consists primarily of plant-based food such as whole grains, fruits, vegetables, nuts, legumes, and seafood, all of which play a multifaced role in keeping our hearts healthy [11]. This healthy diet is an abundant source of vitamins and minerals, dietary fiber, antioxidants, phytochemicals, flavonoids, and other essential nutrients that are inversely associated with cardiovascular disease and all causes of mortality [12]. A suboptimal diet enriched with trans-fats, sugar-sweeteners, refined grains, red meat, salt, and processed foods, on the other hand, is considered a core attributable factor for all causes of mortality and disability. Dietary risk factors would be accountable for 14% of all mortality (7.9 million) and 7% of all disability-adjusted life years (188 million) worldwide [1]. These dietary risk factors are associated with oxidative and inflammatory stress, which are key mechanisms of atherosclerosis development [13].

Earlier, the researcher focused on single and isolated nutrients to examine their effects on health and disease status. This produced misleading and controversial findings regarding the effectiveness of a nutrient synergistically or antagonistically related to the presence of other nutrients and food matrix [14]. Based on this evidence, current dietary recommendations emphasize obtaining a healthy diet rather than restrictions on a single nutrient. For example, in 2015, the American Dietary Guidelines withdrew the rule on cholesterol intake [14]. In that case, the analysis of total dietary patterns would be a practical and effective way of representing the food people eat on a regular basis in real-world settings [12]. Two widely used approaches to identify the dietary patterns of a population are the "Priori approach" and the "Posteriori approach". There are numerous priori approaches (e.g., Dietary Approaches to Stop Hypertension (DASH) diet, Healthy Eating Index (HEI), Mediterranean diet (MED), etc.) to detect dietary patterns based on the intake of predefined food groups [15]. However, the majority of priori dietary pattern related studies were conducted in Western countries, making it challenging to extrapolate the dietary patterns of people residing in less developed countries due to disparate dietary habits, cultures, and availability [16–18]. In contrast, the data-driven posteriori strategy was able to identify the empirical dietary pattern of the population under study, reflecting real-world dietary habits, through statistical methods like factor and cluster analysis of population food consumption data [15,19]. It opens the window of opportunity to assess the dietary pattern and its effect on health and disease state [20].

Bangladesh is undergoing substantial economic development resulting in rapid urbanization, a dietary transition from traditional to Western patterns, and a sedentary lifestyle, which leads to epidemiological transitions [10,21,22]. Non-communicable diseases (NCDs) are now overtripped communicable infectious diseases (68% vs. 11%), with a dramatic rise in cardiovascular-related deaths [23]. Approximately 71% of the population is now exposed to the risk factor of heart disease, which accounts for 17% of all deaths in Bangladesh [24,25]. Several types of heart disease are prevalent among Bangladeshis, including coronary heart disease, cardiomyopathy, congenital heart disease, arrhythmia, rheumatic heart disease, heart failure, and valvular heart disease [10,26]. Among all types, coronary heart disease is the leading cause of mortality and hospitalization [26]. Dietary components (poor consumption of fruits and vegetables, high intake of salt) are the leading contributors to these heart diseases [24], which underscores the urgency of the overall dietary pattern-related risk assessment. Currently, dietary patterns are an emerging field to assess the risk of chronic disease across the world. Multiple studies conducted in Bangladesh found that dietary patterns identified through principal component analysis (PCA) were linked to heart disease [27], CVDs [27], high blood pressure [28], diabetes [29], and skin lesions [30].

However, recent studies suggested that the dietary pattern was region-specific [20,31], and heart disease rates varied widely across the division [32,33]. Many of the studies on heart disease were conducted in hospitals [34–36], had limited sample sizes [9,32,37,38], exhibited statistical issues, and lacked nationwide representation [8]. Only one study assessed the effect of dietary patterns on heart disease in Bangladesh, and this was limited to rural areas [27]. Given the diverse nature of dietary patterns and heart disease in Bangladesh, the use of large-scale country-representative data can help fill this gap. Therefore, the study aimed to evaluate the geospatial variability of dietary patterns and heart disease at both the national and subnational (district) levels and to investigate the association between dietary patterns and heart disease.

## Materials and methods

### Data

This cross-sectional study used data from the Household Income and Expenditure Survey (HIES) 2016/17, a nationally representative survey in Bangladesh. The Bangladesh Bureau of Statistics (BBS) conducted the HIES from April 2016 to March 2017. A two-stage stratified sampling approach was employed to select primary sampling units (PSUs), which are enumeration areas typically encompassing an average of 110 households as utilized in the population and housing census of Bangladesh. At first, Bangladesh was stratified across its eight administrative divisions (Dhaka, Barisal, Chittagong, Mymensingh, Sylhet, Khulna, Rajshahi, and Rangpur). Then, divisions were stratified into its subsequent 64 districts. Each district was further subdivided into smaller geographic units (rural, urban), with the primary division (Dhaka, Chittagong, Khulna, Rajshahi) further subdivided into city corporations and other urban areas. Thus, 132 sub-strata were classified, including 64 urban, 64 rural, and four city corporations. As HIES provides district-level estimates, the sample size was calculated at the district level, where the mean expenditure on household consumption was used as a key indicator. A total of 720 households in each district were determined as sample size, and these households were distributed into 36 PSUs of each district (20 households per PSU). A probability proportion to size (PPS) systematic sampling approach was used to select PSUs in each district. Then, 20 households were selected randomly from a complete household list of each PSU. Thus, a total of 2,304 PSU (36 PSU in every district×64 districts) and 46,080 households (2,304 PSU×20 households per PSU) were the estimated sample size, including 186,207 individuals. All variables were checked thoroughly to identify any missing data. Participants were

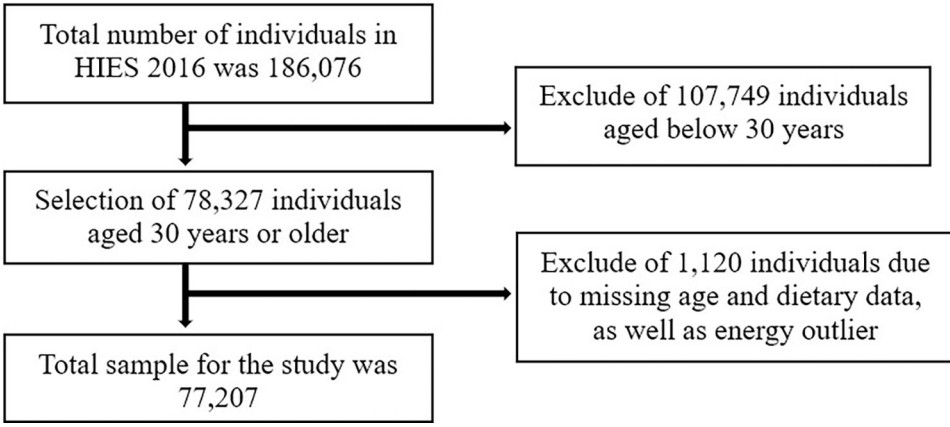

**Fig 1. Flow chart of data extraction from HIES 2016.**

excluded from this study if missing data on any of the major variables—age, disease conditions, dietary information, and energy outlier—was found. However, 77,207 people aged 30 and older were included in this study as the heart disease rate began to rise steeply at this age [2]. The procedure followed to retrieve the sample for this study is presented in the follow-chart (Fig 1). Trained and skilled enumerators collected data using a pretested questionnaire through face-to-face interviews at the participant's household. After the interview and data entry were completed, both the soft and hard copies of the questionnaire were sent to the head-quarters of BBS, where they were extensively examined. The details can be found elsewhere [39].

## Dependent and independent variables of the study

The outcome variable of the study was self-reported heart disease, a dichotomous variable (yes or no). In the HIES survey, module 3 consists of a set of questions related to chronic disease, including chronic heart disease. The respondents were asked whether they suffered from any kind of chronic disease during the previous 12 months or more. If they responded positively, additional inquiries were made to verify the type of chronic disease. The responses were entirely based on the respondent's experiences, signs and symptoms, diagnosis of the disease, and course of treatment for that specific illness [39].

The independent variables included in this study were age (30–39, 40–49, 50–59, and ≥60), sex, marital status (currently married, never married, and ever married), religion (Islam, Hinduism, others), educational status (no education/<primary, primary, secondary and more than secondary), employment status (employed, and unemployed), area of livings (rural, and urban), household size (≤4 members and ≥5 members), and divisions (Mymensingh, Dhaka, Barisal, Khulna, Rangpur, Dinajpur, Chittagong, and Sylhet). There were a few missing data regarding the education and employment status of participants, which was kept missing during further analysis.

## Dietary data processing

The HIES 2016 questionnaire had a distinct section with 133 lines of food items under 17 food groups designed to collect food consumption data over the preceding 14 days through 7 enu-merators' visits. Along with 133 distinct food items, each food group had an entry labeled "Other". This information allowed us to calculate each food type's daily intake (in grams). A

total of 123 food items and "Other" were considered in this study, excluding "tobacco and tobacco products" and "betel leaf and chew goods" items. As dietary data was collected at household levels, individual dietary intake was determined by the adult male equivalent (AME) approach, which is the ratio of the energy requirement of an individual member of a household to the energy requirement of a moderately active adult male aged 18 to 30 years. Individuals were excluded from this study if their daily energy intake was less than 500 Kcal or more than 5,000 Kcal [40]. However, in HIES, the physiological conditions of women were not reported. Hence, all women were assumed to be non-pregnant and non-lactating [41].

## Dietary pattern analysis

All food items were grouped into 27 groups based on similarities in nutritional components and food groups used in earlier studies in Bangladesh [29,39]. Due to the difficulties in identifying specific food items within the "Other" lines during food grouping from "Food grains" and "Vegetables" sections of the questionnaire, the "Other" lines within these sections were considered distinct categories and labeled as "Other grains" and "Other vegetables". Details of food groupings are in the S1 Table. Principal component analysis (PCA) of 27 food groups was used to determine dietary patterns, with the Orthogonal (Varimax) rotation approach employed to get the uncorrelated factors for easier interpretation [20]. In this study, the Kaiser-Meyer-Olkin (KMO) measure of sampling adequacy was 0.81, indicating that the data was eligible for PCA analysis [42]. To determine the number of factors, eigen values>1.30 were used as a cutoff point to retain the factors based on evaluating eigenvalues, scree plots, and interpretability analysis [20]. A total of three factors met the factors mentioned in the retention criteria. Each factor was labeled based on the rotated factor loading of food groups $\geq 0.20$, as these loadings contributed significantly to the dietary pattern [43]. The factor score was derived using the regression method, which provided a score for each factor. The score indicated the participants' adherence to dietary patterns [42]. Finally, the corresponding participants' factor scores of each dietary pattern were categorized into tertiles, where T1 indicated the lowest, and T3 showed the highest adherence or intake of each dietary pattern.

## Statistical analysis

Descriptive statistics (frequency, percentage) of the study variables were performed. Additionally, a bivariate analysis using the Pearson chi-square test for the categorical variable was employed to ascertain the effect of covariates on heart disease across the different regions and to observe variations in the distribution of covariates across the tertile categories of identified dietary patterns. Moreover, the association between self-reported heart disease and dietary patterns was evaluated using bivariate and multivariate logistic regression. In the multivariate model, we adjusted for all potential covariates with p-values less than 0.25 in the bivariate model [44]. These covariates included age, sex, educational status, marital status, religion, employment status, district, energy intake (Kcal), and each dietary pattern. The results of this study were reported as both the crude odds ratios (CORs) and adjusted odds ratios (AORs) with 95% confidence intervals (CIs). Model fitness was checked by the Hosmer-Lemeshow goodness-of-fit test (p = 0.88). Prior to conducting multivariate logistic regression, multicollinearity through variance inflation factors (VIF) was examined, which showed VIF values below 3, indicating the absence of collinearity issues [45]. All statistical significance tests were based on $p < 0.001$, $p < 0.01$, and $p < 0.05$. These analyses were carried out in STATA 14.2 (StataCorp, USA), where the 'svy' command was used to adjust cluster, strata, and sampling weights.

## Spatial analysis

The prevalence of self-reported heart disease and the median factor score of each identified dietary pattern at district levels were used in the spatial analysis. These values were exported to the district shapefile of the Bangladesh map using Arc GIS 10.8.1, where geospatial analysis (thematic map, global autocorrelation analysis) was conducted [46]. Spatial Autocorrelation (Global Moran's I) was used in this study to determine whether any significant spatial clustering of the prevalence of self-reported heart disease and adherence to identified dietary patterns exists [47]. The spatial analysis was conducted in Arc GIS 10.8.1.

## Ethical considerations

Since the de-identified data for this study came from secondary sources, it did not require ethical approval. However, ethical approval was obtained for the HIES, and details can be found elsewhere [39].

## Results

### Sociodemographic characteristics of the participants

A total of 77,207 participants aged 30 years and older were included in this study, with roughly equal numbers of males and females. Most of the participants were less than 60 years old (81.1%). The majority of the participants were married (88.5%), had no education or less than primary level education (48.4%), practiced Islam (87.9%), lived in rural settings (73.1%) and had ≤4 family members in the household (55.6%). Among the total participants, about half of them were employed (50.3%) (Table 1).

### Distribution of the prevalence of heart disease

The overall prevalence of self-reported heart disease among 30 years and older was 3.6% (Fig 2A). However, this study showed that the distribution of heart disease was significantly clustered at the regional level, as indicated by the global Moran's I index value which was 0.35 (p <0.001) (Fig 2B). Prevalence of heart disease varied both at the divisional level, from 5.2% in Khulna to 1.7% in Mymensingh (Fig 2A) and the district level, from 0.6% to 10.4% (Fig 2C). The high prevalence of heart disease was clustered mainly in the western-south, some southern coastal areas, and peripheral areas of the eastern region, while the low prevalence was observed in the northern, central-north, and southern-east hill tracts region (Fig 2C).

However, nationally, this study did not find any significant gender difference in heart disease rate; however, significantly high prevalence was found among males in Barisal, Khulna, and Mymensingh. Heart disease rate increased significantly with advancing age. Moreover, a significantly high prevalence was observed among participants who were ever-married, unemployed, and adherent to Islam faith. The overall heart disease rate was high among people practicing Islam; however, in Dhaka, the rate was significantly high among other religions. Residential area proved to be a crucial factor in Rangpur, as the prevalence was significantly high in urban areas (Table 2).

### Characteristics of dietary patterns

The PCA analysis identified three dietary patterns, which explained 25.2% of the total dietary intake variance cumulatively. The dietary patterns were labeled based on the factor loading of the food groups on the respective dietary pattern. The "festival pattern" explained 10.7% of the variance, characterized by a higher intake of pulses, fruits, processed rice, dairy, sugar and sweetmeats, fats and oil, spices, meat, and carbonated drinks usually consumed on a festive

**Table 1. Sociodemographic characteristics of the study participants (N = 77,207).**

| Variables | Frequency (n) | Percentage (%) |
|---|---|---|
| **Sex** | | |
| Male | 38,918 | 50.6 |
| Female | 38,289 | 49.4 |
| **Age (in years)** | | |
| 30–39 | 27,787 | 36.7 |
| 39–49 | 20,536 | 26.5 |
| 50–59 | 14,018 | 17.9 |
| ≥60 | 14,866 | 18.9 |
| **Education levels** | | |
| No education / <primary | 36,915 | 48.4 |
| Primary | 17,707 | 22.3 |
| Secondary | 17,127 | 22.3 |
| More than secondary | 5,317 | 7.0 |
| **Marital status** | | |
| Currently married | 68,494 | 88.5 |
| Never married | 1,028 | 1.3 |
| Ever married | 7,685 | 10.2 |
| **Religion** | | |
| Islam | 66,008 | 87.9 |
| Hinduism | 8,611 | 10.7 |
| Others | 2,588 | 1.4 |
| **Employment status** | | |
| Employed | 39,118 | 50.3 |
| Unemployed | 38,083 | 49.7 |
| **Area of living** | | |
| Rural | 53,864 | 73.1 |
| Urban | 23,343 | 26.9 |
| **Household size** | | |
| ≤4 members | 43,961 | 55.6 |
| ≥5 members | 33,246 | 44.4 |
| **Divisions** | | |
| Barisal | 7,827 | 6.3 |
| Chittagong | 12,849 | 19.8 |
| Dhaka | 15,365 | 23.8 |
| Khulna | 12,490 | 11.4 |
| Mymensingh | 4,652 | 7.4 |
| Rajshahi | 9,628 | 12.9 |
| Rangpur | 9,390 | 10.9 |
| Sylhet | 5,006 | 7.5 |

day in Bangladesh. The second dietary pattern was labeled as the "pickles and fast foods patterns", which was highly loaded with pickles, fast foods, non-carbonated drinks, deep-fried snacks, biscuits/prepared foods, wheat/flour, non-starchy vegetables, fish, and jam jelly, explained 7% of the total variance. The third dietary pattern was positively loaded with rice, starchy vegetables, salt, leafy vegetables, non-starchy vegetables, and spices, which were labeled as "rice and vegetable patterns", accounting for 7.5% of the variance (Table 3).

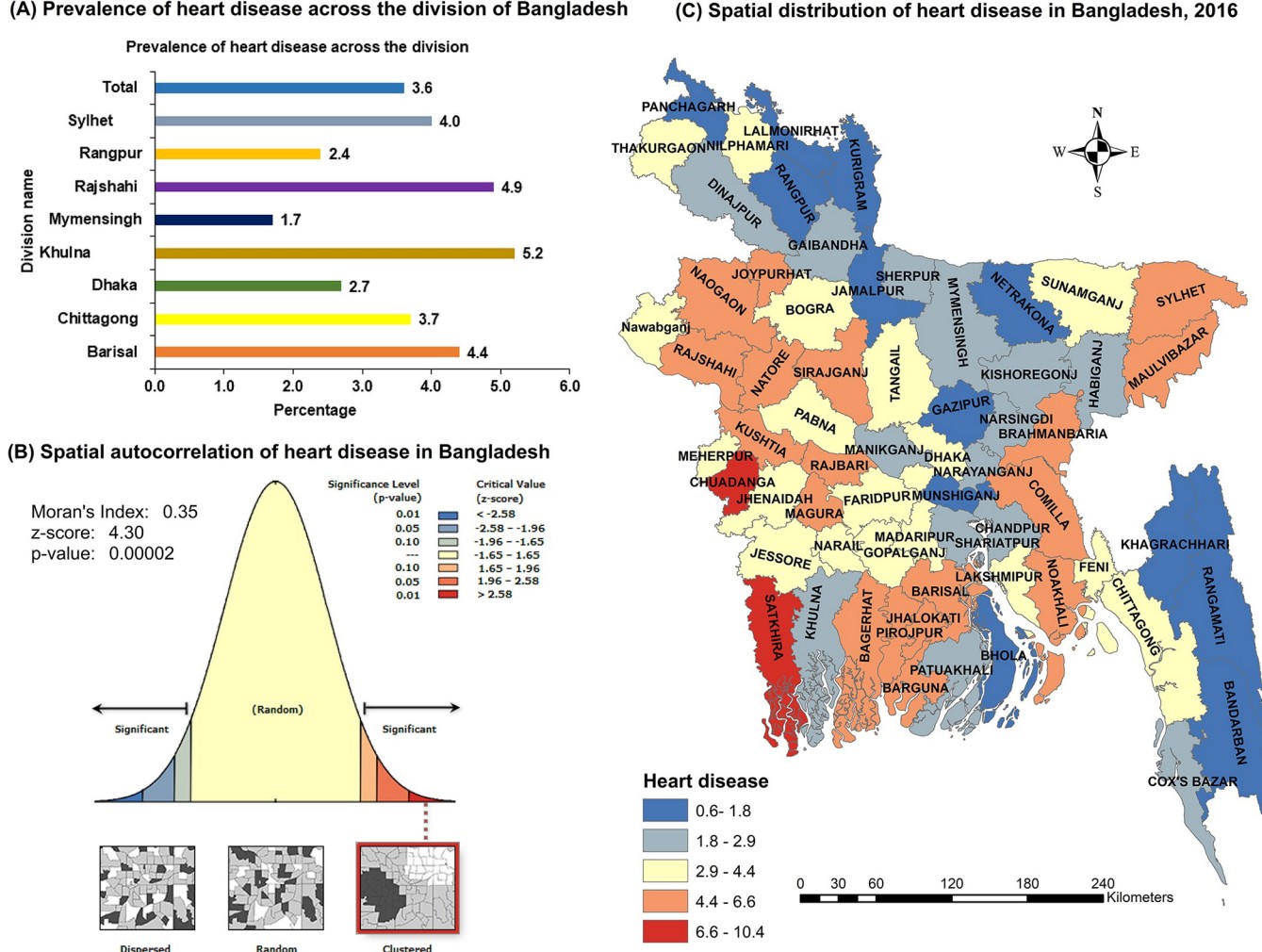

**Fig 2. Spatial variation of self-reported heart disease among people aged 30 years and older.** Spatial distribution of self-reported heart disease in Bangladesh from HIES 2016. (A) The prevalence of self-reported heart disease at division levels in Bangladesh. (B) The spatial autocorrelation of heart disease in Bangladesh. (C) In the map, the red color indicates a higher prevalence, the yellow color indicates moderate prevalence, and the blue indicates a lower prevalence of heart disease.

Table 4 shows the background characteristics of the participants by tertile categories of dietary patterns. Accordingly, individuals with the highest tertile of the "festival pattern" were significantly more likely to be male, aged 30 to 39 years, had completed more than secondary education, never married, employed persons, had ≤4 members in a family, living in the urban areas and be located in the Chittagong division when compared to the individuals in the lowest tertile of this pattern. In the case of the "pickles and fast foods patterns", the highest score was significantly adopted by males, aged 30 to 39 years, had more than secondary education, were never married, employed, ≤4 members in the family, lived in the urban area and located in Chittagong division when compared to those who were adopted to the lowest score. In addition, participants with the top tertile of "rice and vegetable pattern" were significantly more likely to be male, aged 50 to 59 years, had primary education, never married, practiced Hinduism, were employed, ≤4 members in the family, lived in rural areas, and located in Rangpur division than those with the lowest tertile.

**Table 2. Prevalence of self-reported heart disease among 30 years and older people by sample characteristics and divisions of Bangladesh.**

| Covariates† | Barisal, n = 7,827 | Chittagong, n = 12,849 | Dhaka, n = 15,365 | Khulna, n = 12,490 | Mymen-singh, n = 4,652 | Rajshahi, n = 9,628 | Rangpur, n = 9,390 | Sylhet, n = 5,006 | Total, N = 77,207 |
|---|---|---|---|---|---|---|---|---|---|
| **Sex** | | | | | | | | | |
| Male | 3.8** | 4.0 | 2.5 | 4.8* | 1.3* | 4.5 | 2.5 | 3.9 | 3.4 |
| Female | 5.1 | 3.4 | 2.8 | 5.7 | 2.2 | 5.2 | 2.3 | 4.0 | 3.7 |
| **Age (in years)** | | | | | | | | | |
| 30–39 | 2.5*** | 1.6*** | 1.1*** | 2.9*** | 1.1* | 3.2*** | 1.7** | 1.8*** | 1.9*** |
| 39–49 | 4.2 | 3.0 | 1.9 | 5.0 | 1.1 | 3.8 | 2.2 | 3.6 | 3.0 |
| 50–59 | 6.0 | 6.8 | 4.3 | 6.5 | 2.3 | 4.9 | 2.8 | 5.4 | 5.0 |
| ≥60 | 6.4 | 6.0 | 5.8 | 8.7 | 3.0 | 9.4 | 3.6 | 7.4 | 6.3 |
| **Education levels** | | | | | | | | | |
| No education / <primary | 3.6* | 3.8 | 2.6 | 6.0*** | 1.4 | 5.0 | 2.2 | 4.0 | 3.5 |
| Primary | 5.3 | 3.5 | 2.3 | 5.8 | 2.7 | 5.1 | 2.8 | 4.1 | 3.8 |
| Secondary | 4.8 | 3.9 | 2.7 | 3.7 | 1.8 | 4.3 | 2.6 | 4.1 | 3.4 |
| More than secondary | 3.7 | 3.7 | 3.9 | 3.6 | 1.9 | 4.5 | 2.3 | 1.7 | 3.6 |
| **Marital status** | | | | | | | | | |
| Currently married | 4.3 | 3.6 | 2.5* | 5.0* | 1.4** | 4.7 | 2.4 | 3.7 | 3.4*** |
| Never married | 2.3 | 1.8 | 1.1 | 3.6 | 1.9 | 2.8 | 0 | 0 | 1.6 |
| Ever married | 6.1 | 4.9 | 4.4 | 7.5 | 3.9 | 6.5 | 2.3 | 5.6 | 5.9 |
| **Religion** | | | | | | | | | |
| Islam | 4.3 | 4.0* | 2.6** | 5.4 | 1.7 | 4.9 | 2.5 | 4.3 | 3.6* |
| Hinduism | 5.3 | 3.7 | 3.1 | 4.0 | 2.3 | 4.2 | 2.1 | 2.7 | 3.3 |
| Others | 5.5 | 0.3 | 21.7 | 10.0 | 0 | 0.9 | 0 | 1.2 | 1.1 |
| **Employment status** | | | | | | | | | |
| Employed | 3.1*** | 3.1 | 2.1** | 4.4 | 1.2* | 3.4*** | 2.2 | 2.9** | 2.8*** |
| Unemployed | 5.5 | 4.2 | 3.3 | 6.1 | 2.3 | 6.5 | 2.7 | 4.8 | 4.4 |
| **Area of living** | | | | | | | | | |
| Rural | 4.2 | 3.3 | 2.7 | 5.3 | 1.6 | 4.8 | 2.2* | 4.0 | 3.5 |
| Urban | 5.4 | 4.6 | 2.6 | 4.8 | 2.4 | 5.0 | 3.5 | 3.6 | 3.7 |
| **Household size** | | | | | | | | | |
| ≤4 members | 4.3 | 4.2 | 2.6 | 5.3 | 1.5 | 4.8 | 2.6 | 3.7 | 3.6 |
| ≥5 members | 4.5 | 3.4 | 2.8 | 5.0 | 2.0 | 4.9 | 2.1 | 4.1 | 3.5 |

*p <0.05

**p < 0.01

***p <0.001, from the chi-square test.

† The values were reported as row percentages.

## Spatial distribution of dietary patterns

The distribution of dietary patterns varied greatly across the regions in Bangladesh. The global Moran's I index values for the "festival pattern", "pickles and fast foods pattern", and "rice and vegetable pattern" were 0.51, 0.32, and 0.43 respectively at $p < 0.001$, indicating significant clustering of each of the dietary patterns at spatial levels in Bangladesh (Fig 3).

The district-wise adherence to the "festival pattern", "pickles and fast foods pattern", and "rice and vegetable pattern" was shown in Fig 4. Individuals in the northern and western regions were less likely to adhere to the "festival pattern" which was more prevalent in the

**Table 3. Factor-loading matrix of the three major dietary patterns.**

| Variable* | Festival pattern | Pickles and fast food pattern | Rice and vegetable pattern |
|---|---|---|---|
| Rice | | | 0.492 |
| Processed rice | 0.355 | -0.202 | |
| Biscuits/prepared foods | | 0.289 | |
| Pulse/legumes | 0.365 | | |
| Wheat or flour | | 0.266 | |
| Fish | | 0.219 | |
| Meat | 0.250 | | |
| Fruits | 0.358 | | |
| Non-starchy vegetables | | 0.256 | 0.278 |
| Starchy vegetables | | | 0.460 |
| Leafy vegetables | | | 0.316 |
| Dairy | 0.332 | | |
| Pickles | | 0.377 | |
| Fast foods | | 0.372 | |
| Jam and jelly | | 0.207 | |
| Deep-fried snacks | | 0.314 | |
| Fats and oil | 0.296 | | |
| Salt | | | 0.405 |
| Spices | 0.259 | | 0.250 |
| Carbonated drinks | 0.250 | | |
| Non-carbonated drinks | | 0.351 | |
| Sugar and sweetmeat | 0.325 | | |
| Variance explained (%) | 10.7% | 7.0% | 7.5% |

* Items that were also included but did not load on any pattern were other grains, eggs, other vegetables, sauce and vinegar, sirka, and outside meal. Only food groups having factor loadings >0.200 were shown.

eastern and central areas (Fig 4A). The "pickles and fast foods pattern" was concentrated in western-south, southern, and eastern regions while it was less commonly adopted by individuals in the northern-west area (Fig 4B). In addition, residents in the north and western-south regions were more likely to adhere to the "rice and vegetable pattern" (Fig 4C).

## Association between heart disease and dietary patterns

Table 5 represents the multivariate logistic regression model of the association of self-reported heart disease with dietary patterns. In the adjusted model, the highest tertile of the "pickles and fast foods pattern" exhibited a 50% higher risk of developing heart disease in comparison to those in the lowest tertile (AOR: 1.50, 95% CI: 1.27–1.76, p <0.001). Conversely, the highest adherence to the "rice and vegetable pattern" showed a 22% lower likelihood of developing heart disease when compared to those with lower adherence (AOR: 0.78, 95% CI: 0.64–0.95, p <0.05). In the case of the "festival pattern", however, no significant link with the risk of developing heart disease was identified.

## Discussion

Dietary factors can play a significant role in either promoting or aggravating the pathophysiology of heart disease. This study explored the dietary patterns and risk of heart disease in Bangladesh. Using principal component analysis, this study identified three major dietary

**Table 4. Characteristics of the study participants by tertile categories of dietary patterns.**

| Characteristics† | Festival pattern | | | | Pickles and fast food pattern | | | | Rice and vegetable pattern | | | |
|---|---|---|---|---|---|---|---|---|---|---|---|---|
| | T1 | T2 | T3 | p-value* | T1 | T2 | T3 | p-value* | T1 | T2 | T3 | p-value* |
| **Sex** | | | | | | | | | | | | |
| Male | 25.3 | 33.2 | 41.5 | <0.001 | 25.6 | 31.3 | 43.1 | <0.001 | 23.5 | 34.0 | 42.5 | <0.001 |
| Female | 37.0 | 34.3 | 28.7 | | 36.5 | 33.8 | 29.7 | | 46.0 | 33.4 | 20.6 | |
| **Age (in years)** | | | | | | | | | | | | |
| 30–39 | 28.6 | 33.1 | 38.3 | <0.001 | 27.8 | 32.2 | 40.0 | <0.001 | 31.8 | 34.8 | 33.4 | <0.001 |
| 39–49 | 31.4 | 34.8 | 33.8 | | 30.8 | 32.6 | 36.6 | | 33.7 | 34.2 | 32.1 | |
| 50–59 | 30.2 | 33.0 | 36.8 | | 30.4 | 31.8 | 37.8 | | 30.1 | 34.5 | 35.4 | |
| ≥60 | 36.0 | 34.3 | 29.7 | | 38.1 | 33.5 | 28.4 | | 45.7 | 30.1 | 24.2 | |
| **Education levels** | | | | | | | | | | | | |
| No education / <primary | 39.7 | 34.3 | 26.0 | <0.001 | 39.0 | 33.2 | 27.8 | <0.001 | 34.2 | 33.9 | 31.9 | <0.001 |
| Primary | 29.1 | 35.0 | 35.9 | | 29.1 | 33.5 | 37.4 | | 32.9 | 34.4 | 32.7 | |
| Secondary | 20.9 | 33.3 | 45.8 | | 21.5 | 31.8 | 46.7 | | 35.6 | 33.5 | 30.9 | |
| More than secondary | 9.8 | 27.0 | 63.2 | | 11.4 | 26.7 | 61.9 | | 40.2 | 30.4 | 29.4 | |
| **Marital status** | | | | | | | | | | | | |
| Currently married | 30.3 | 33.8 | 35.9 | <0.001 | 30.1 | 32.5 | 37.4 | <0.001 | 33.0 | 34.3 | 32.7 | <0.001 |
| Never married | 23.1 | 35.0 | 41.9 | | 24.7 | 32.5 | 42.8 | | 33.8 | 26.4 | 39.8 | |
| Ever married | 38.7 | 33.1 | 28.2 | | 39.5 | 32.9 | 27.6 | | 49.1 | 28.9 | 22.0 | |
| **Religion** | | | | | | | | | | | | |
| Islam | 30.8 | 33.6 | 35.6 | 0.533 | 30.3 | 32.5 | 37.2 | 0.249 | 35.1 | 33.8 | 31.1 | 0.009 |
| Hinduism | 32.1 | 34.8 | 33.1 | | 36.0 | 33.3 | 30.7 | | 31.1 | 32.9 | 36.0 | |
| Others | 38.5 | 32.1 | 29.4 | | 31.0 | 29.9 | 39.1 | | 33.7 | 32.0 | 34.3 | |
| **Employment status** | | | | | | | | | | | | |
| Employed | 26.6 | 33.0 | 40.4 | <0.001 | 26.2 | 31.1 | 42.7 | <0.001 | 23.3 | 34.0 | 42.7 | <0.001 |
| Unemployed | 35.6 | 34.5 | 29.9 | | 35.8 | 33.9 | 30.3 | | 46.1 | 33.4 | 20.5 | |
| **Household size** | | | | | | | | | | | | |
| ≤4 members | 26.2 | 32.7 | 41.1 | <0.001 | 27.3 | 31.2 | 41.5 | <0.001 | 24.1 | 33.9 | 42.0 | <0.001 |
| ≥5 members | 37.1 | 35.1 | 27.8 | | 35.6 | 34.2 | 30.2 | | 47.8 | 33.5 | 18.7 | |
| **Area of livings** | | | | | | | | | | | | |
| Rural | 34.7 | 33.3 | 32.0 | <0.001 | 35.9 | 33.1 | 31.0 | <0.001 | 32.1 | 33.5 | 34.4 | <0.001 |
| Urban | 21.1 | 34.9 | 44.0 | | 17.7 | 30.9 | 51.4 | | 41.7 | 34.1 | 24.2 | |
| **Divisions** | | | | | | | | | | | | |
| Barisal | 27.7 | 37.0 | 35.3 | <0.001 | 35.3 | 35.7 | 29.0 | <0.001 | 49.7 | 31.4 | 18.9 | <0.001 |
| Chittagong | 15.8 | 34.7 | 49.5 | | 14.7 | 27.7 | 57.6 | | 47.9 | 32.2 | 19.9 | |
| Dhaka | 21.5 | 33.8 | 44.7 | | 24.1 | 33.0 | 42.9 | | 34.5 | 34.9 | 30.6 | |
| Khulna | 38.3 | 36.9 | 24.8 | | 22.4 | 40.9 | 36.7 | | 24.1 | 37.9 | 38.0 | |
| Mymensingh | 41.0 | 32.8 | 26.2 | | 43.4 | 35.8 | 20.8 | | 32.2 | 36.6 | 31.2 | |
| Rajshahi | 38.1 | 34.8 | 27.1 | | 37.9 | 34.3 | 27.8 | | 32.1 | 33.2 | 34.7 | |
| Rangpur | 57.6 | 24.2 | 18.2 | | 60.0 | 25.0 | 15.0 | | 17.9 | 26.7 | 55.4 | |
| Sylhet | 32.8 | 36.3 | 30.9 | | 38.7 | 33.0 | 28.3 | | 34.6 | 37.6 | 27.8 | |

T, tertiles.

* P values were based on the χ2 test to detect differences among the tertile categories of dietary patterns.

† The values of all categorical variables were reported as percentages.

**(A) Spatial autocorrelation of festival pattern in Bangladesh, 2016**

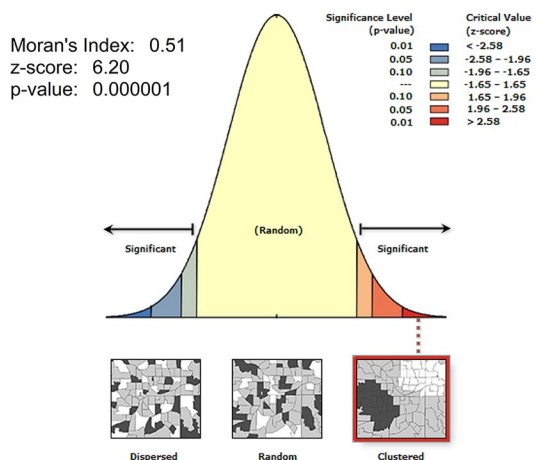

**(C) Spatial autocorrelation of rice and vegetable pattern in Bangladesh, 2016**

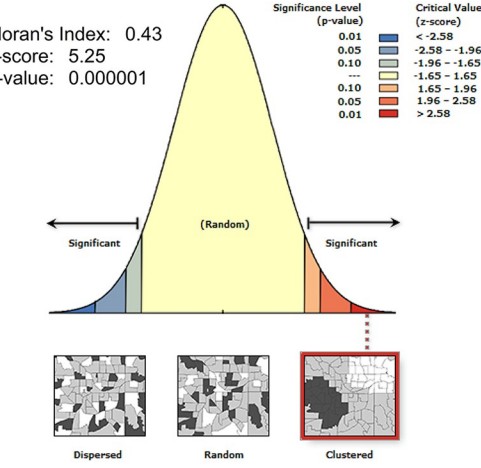

**(B) Spatial autocorrelation of pickles and fast foods pattern in Bangladesh, 2016**

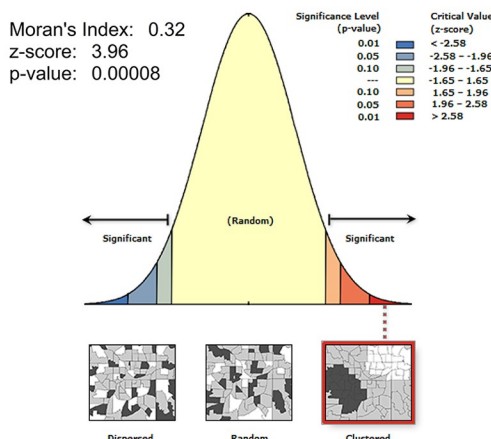

**Fig 3. The spatial autocorrelation of dietary patterns among 30 years and older.**

patterns: "festival pattern", "pickles and fast foods pattern", and "rice and vegetable pattern". The "pickles and fast foods pattern" was positively associated with heart disease, while an inverse association was found with the "rice and vegetable pattern". However, "festival pattern" was not significantly associated with heart disease. Our study revealed spatial dependencies in both the prevalence of heart disease and adherence to dietary patterns and observed cluster patterns.

This study found that the overall prevalence of self-reported heart disease was 3.6%, which was almost consistent with prior studies conducted in Bangladesh [32,37,38] and India [48]. However, our study found a comparatively lower prevalence of heart disease when compared to the other studies in Bangladesh [9,33]. The high prevalence might be due to the methodological differences (study design, study settings, subject of the study) and time variation of the study conducted. Consistent with the previous studies, the heart disease rate increased with advancing age, and a higher prevalence was also observed among the unemployed [32,49,50]. This is expected as the aging process inherently leads to a decline in cardiovascular function, which is a non-modifiable risk factor for heart disease [51]. Besides, unemployed persons had poor access to various facilities and services, which may induce heart disease risk [49].

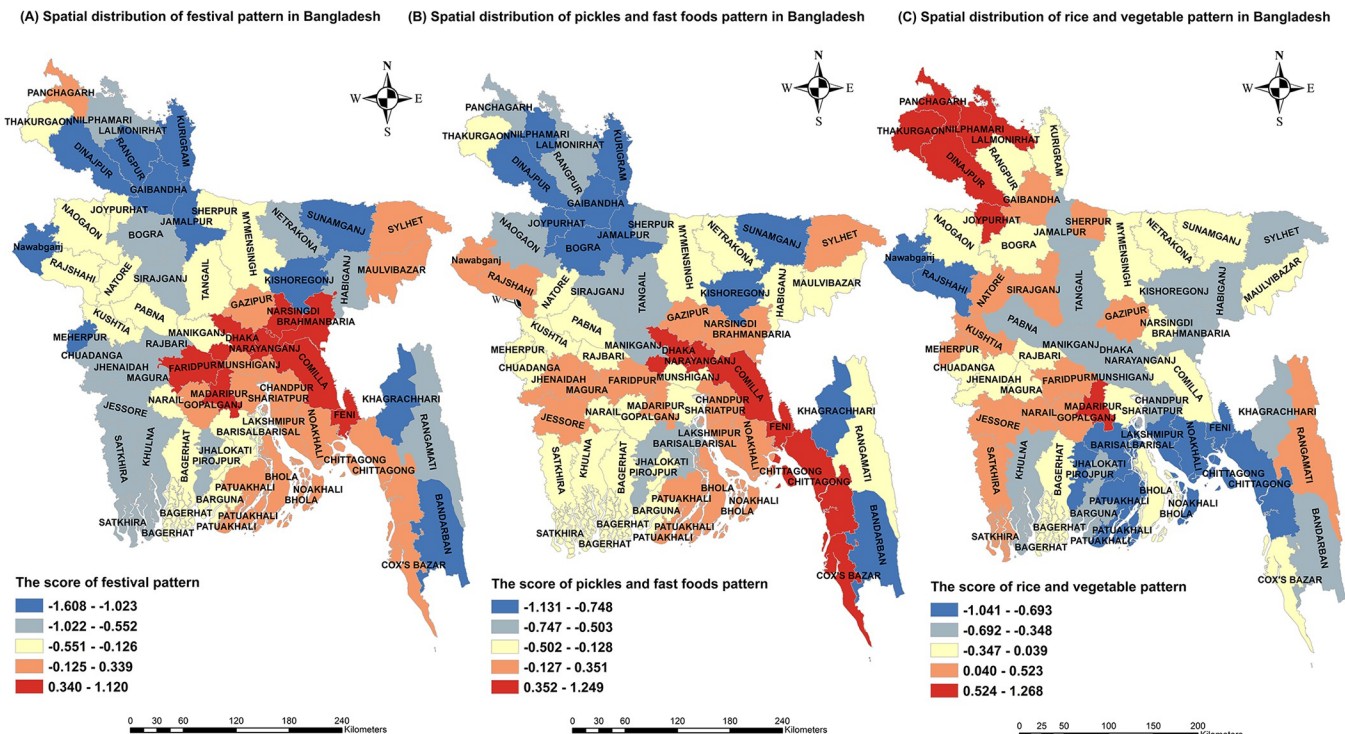

**Fig 4. Spatial variation of the dietary pattern.** Spatial distribution of dietary patterns in Bangladesh from HIES 2016. The adherence to the (A) "festival pattern", (B) "pickles and fast foods pattern", and (C) "rice and vegetable pattern" were presented at district levels in Bangladesh. On the map, red indicates higher adherence, yellow indicates moderate adherence, and blue indicates a lower adherence to each dietary pattern.

**Table 5. Association between self-reported heart disease across the tertile categories of dietary patterns.**

| Dietary patterns | COR (95% CI) | AOR (95% CI)[a] |
|---|---|---|
| **Festival pattern** | | |
| T1 (Lowest) | Ref | Ref |
| T2 | 1.10 (0.96–1.26) | 1.02 (0.90–1.17) |
| T3 (Highest) | 1.18 (1.04–1.35) * | 1.12 (0.96–1.32) |
| **Pickles and fast foods pattern** | | |
| T1 (Lowest) | Ref | Ref |
| T2 | 1.24 (1.09–1.42) ** | 1.26 (1.10–1.44) ** |
| T3 (Highest) | 1.36 (1.18–1.57) *** | 1.50 (1.27–1.76) *** |
| **Rice and vegetable pattern** | | |
| T1 (Lowest) | Ref | Ref |
| T2 | 0.88 (0.78–1.00) * | 0.94 (0.81–1.08) |
| T3 (Highest) | 0.76 (0.66–0.86) *** | 0.78 (0.64–0.95) * |

* $p < 0.05$

**$p < 0.01$

***$p < 0.001$.

AOR, adjusted odds ratio; COR, crude odds ratio; CI, confidence interval; T, tertiles of each dietary pattern score.

[a]Adjusted for age, sex, educational status, marital status, religion, employment status, district, energy intake (Kcal), and each dietary pattern.

Residential location emerged as a crucial factor in our present study, as it revealed a substantial variation in self-reported heart disease at district and division levels that align with findings from other studies [32,33]. This could be attributed to different dietary patterns, environmental factors, culture, food habits, and lifestyles prevalent in different regions.

In line with an earlier study conducted in Bangladesh, the present study exhibited a wide variation and was a region-specific dietary pattern [31]. This phenomenon might be ascribed to the apparent differences in dietary practices across the nation, stemming from regional disparities in factors such as poverty, food insecurity, agricultural production, rainfall patterns, riverbank erosion, climate change, and natural catastrophes [52]. Our study found that in the north and western-south regions, people mainly adhered to the "rice and vegetable pattern"; in the eastern and central areas, people mostly adhered to the "festival pattern"; and the "pickles and fast foods pattern" was mostly adhered to the western-south, southern and eastern regions. A recent study on the cost of a recommended diet identified that the expense of adhering to the recommended intake of cereals, fruits, and vegetables was the lowest in the northern and southwestern regions of Bangladesh. However, these areas also faced higher levels of unaffordability in achieving a recommended diet compared to other parts of the country [53]. Therefore, people living in north Bengal mainly survive on low-cost foods like starches and vegetables, which were mirrored in rice, as well as low-diversity patterns, traditional patterns, and monotonous starch-based diets. In contrast, across the southern region, meat and fish patterns, high dietary diversity, and modern dietary patterns with a higher intake of processed foods were dominant [20,31,54]. This diversity could be a result of different socioeconomic conditions. North Bengal is a region plagued by poverty, vulnerable to drought, with severe food insecurity and reliance on subsistence agriculture. Compared to the north, the central, southeastern, and eastern regions had a lower rate of poverty and food insecurity and a higher standard of living [54]. Dietary intake is greatly influenced by socioeconomic conditions where cereal grains make up the bulk of the poverty-stricken people's diet, and processed foods, restaurant foods, and animal-sourced foods are more likely to be consumed by city dwellers and affluent people [55]. As a result, both the "festival pattern" and the "pickles and fast foods pattern" were lower in the northern region, especially the northern-west, as these patterns primarily consisted of animal-derived and processed food items.

The present study identified that the "pickles and fast foods pattern" (pickles, fast foods, jam and jelly, deep-fried snacks, wheat, or flour) increased the heart disease risk by 50%. Conversely, adherence to the "rice and vegetable pattern" (rice, starchy vegetables, leafy vegetables, non-starchy vegetables, and spices) showed a reduced risk of heart disease as this pattern lowered the risk by 22%. Likewise, in Bangladesh, several studies suggested that higher consumption of red meat, eggs, fast food, soft drinks, and food rich in salt increased the risk of CVDs. In contrast, consumption of fruits and vegetables was associated with a lower risk, which was reflected in our study [27–30,56]. Fast food, deep-fried, and sugary snacks usually contain a higher amount of refined carbohydrates, sugars, fats, saturated fats, cholesterols, and lower micronutrients; all of these ingredients are known to be atherogenic [57,58]. These foods are usually prepared in an unhealthy way, such as deep frying in oil, reusing oil, or cooking for an extended period, resulting in increased trans-fat, which negatively affects lipid metabolism, increased inflammatory and oxidative stress, impaired endoplasmic function, apoptosis, and autophagy [59]. Besides, lower micronutrients, particularly vitamin B, are linked to hyperhomocysteinemia, a well-known risk factor for coronary heart disease [8]. In addition to the high sugar content, refined grains with a high glycemic load can elevate inflammatory markers, resulting in the development of atherosclerosis. These high glycemic foods are also responsible for type 2 diabetes and obesity, which are the established risk factors for heart disease [60]. Conversely, vegetables and spices contain numerous bioactive substances (vitamins, minerals,

fibers, phytonutrients) which have been shown to have a cardioprotective role through a variety of mechanisms, including regulating blood pressure, insulin sensitivity, and lipid metabolism; anti-oxidation, anti-inflammatory, and anti-platelet properties; better endothelial function, reducing myocardial damage, modulation of enzymatic activities, gene expressions and other signaling pathways [61,62]. Furthermore, those who consume more fruits and vegetables usually lead a healthy lifestyle with more physical activity and less smoking, which may result in a reduced likelihood of heart disease [63]. Adequate fruits and vegetables consumption could lower the global burden of ischemic heart disease by 31% [64]. Hence, the dietary guidelines should prioritize the adoption of healthy eating practices such as Mediterranean dietary patterns, Prudent patterns, DASH diet, and plant-based dietary patterns. These emphasize the consumption of essential healthy food items such as fruits and vegetables, whole grains, legumes, fish, and poultry while discouraging unhealthy eating practices seen in Western dietary patterns characterized by higher intake of refined grains, red meat, fast foods, processed foods, and sugar [12].

However, the "festival pattern" was not significantly associated with heart disease. Though this pattern was loaded with some unhealthy food items such as sugar, sweetmeat, and carbonated drinks, it was mainly represented by pulses/legumes and fruits. It is established that sugar and carbonated beverages increase the risk of heart disease, but plant-based items such as fruits and pulses offer protection against heart disease [11]. It appears that in the "festival pattern", the effect of the healthy food intake masked the impact of unhealthy food intake, resulting in no significant association.

In this study, a high rate of heart disease prevalent in the western-south, southern, central, and eastern regions could be due to more adherence to the "festival pattern" and "pickles and fast foods pattern", which consisted of unhealthy food components and this higher rate aligns with the findings of the previous study [33]. Conversely, the beneficial effect of adherence to the "rice and vegetable pattern" was exhibited in a lower rate of heart disease in Bangladesh's north, central-north, and southern-east hilly tracts region. The rate was in line with an earlier study [33]. Though in the western-south "rice and vegetable pattern" also prevailed, its protective effect might be masked by the "pickles and fast foods pattern" as well as factors such as the salinity of water and the presence of arsenic content, which increased the risk of CVD in the coastal region [30,65,66].

The rising trend of NCDs in Bangladesh is a significant obstacle to achieving sustainable development goals. Heart disease represents a significant health burden in Bangladesh, a challenge exacerbated by the country's limited healthcare infrastructure, a shortage of skilled personnel, inadequate epidemiological data, and widespread poverty [22,67]. Bangladesh has already adopted the multisectoral plan to reduce a relative 25% of NCD-related mortality, where heart disease is a significant contributor [68]. Achieving this goal would be unattainable without achieving healthy dietary practices. This study identified the prevailing dietary patterns across the country, providing valuable insights for the government to develop effective guidelines for tailoring heart disease prevention programs. Our study recommends a public awareness campaign to promote a heart-healthy dietary choice, such as incorporating rice, vegetables, fruits, and spices, while discouraging the consumption of fast food and sugary items through mass media, educational institutions, and other channels. Furthermore, the government should impose a tax on unhealthy food items and provide subsidies on heart-healthy food items to ensure affordability for the general population. As this study identified the vulnerable regions to heart disease and unhealthy dietary patterns, it can provide valuable insights for policymakers for tailoring region-specific interventions.

The main strength of the present study was its large country-representative sample size. Furthermore, use of sampling weight allowed us to extrapolate the findings to the entire

population of Bangladesh. Additionally, the spatial distribution of heart disease and dietary patterns across the district was another strength of this study. Nonetheless, this study has some limitations that warrant consideration in future research. The HIES did not include specific details on the method used for heart disease diagnosis. Additionally, due to the absence of intra-household food distribution data, the individual dietary intake was determined using the AME approach which has inherent limitations [69]. Furthermore, the factor analysis involved making arbitrary decisions regarding food grouping, rotation method, eigenvalue and scree plot analysis, and a cutoff value of factor loadings, all of which may influence the results and interpretation [70]. In addition, the absence of data on potential confounders such as body mass index, smoking status, and physical activity posed limitations in adjusting for these variables in the multivariate analysis.

## Conclusion

This study highlights the significant impact of dietary patterns on heart disease risk in Bangladesh, uncovering notable geospatial variations. A total of three primary dietary patterns was identified in Bangladesh- "festival pattern", "pickles and fast foods pattern", and "rice and vegetable pattern". The "pickles and fast foods pattern" was associated with increased risk of heart disease, while the "rice and vegetable pattern" demonstrated a protective effect. Regions with higher adherence to the "festival pattern", and "pickles and fast foods pattern", such as the western-south, southern, central, and eastern areas, had higher heart disease rates, whereas areas like the north and central-north, adhering more to the "rice and vegetable pattern," had lower rates. These findings underscore the need for public health initiatives promoting heart-healthy diets, advocating for public awareness campaigns, taxing unhealthy foods, and subsidizing nutritious options, with region-specific interventions to target vulnerable areas, especially in the western-south, southern, central, and eastern regions of Bangladesh. Nevertheless, these findings must be supported by additional research to establish the link between dietary patterns and heart disease conclusively.

## Supporting information

**S1 Table. Food items and food groupings.**
(PDF)

## Acknowledgments

The present study was carried out utilizing data from the Bangladesh Household Income and Expenditure Survey 2016–17. We would like to express our gratitude to the BBS and participants of HIES.

## Author Contributions

**Conceptualization:** Rafid Hassan, Md. Ruhul Amin.

**Data curation:** Rafid Hassan, Masum Ali.

**Formal analysis:** Rafid Hassan.

**Funding acquisition:** Rafid Hassan, Md. Ruhul Amin.

**Methodology:** Rafid Hassan, Md. Ruhul Amin.

**Software:** Rafid Hassan.

**Supervision:** Md. Ruhul Amin.

**Writing – original draft:** Rafid Hassan, Masum Ali, Md. Ruhul Amin.

**Writing – review & editing:** Rafid Hassan, Masum Ali, Sanjib Saha, Sadika Akhter, Md. Ruhul Amin.

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
