## [Decision Letter · Decision Letter 0]

26 Sep 2023

PONE-D-23-21261Geospatial variation in dietary patterns and their association with heart disease in Bangladeshi population: Evidence from a nation-wide surveyPLOS ONE

Dear Dr. Amin,

Thank you for submitting your manuscript to PLOS ONE. After careful consideration, we feel that it has merit but does not fully meet PLOS ONE’s publication criteria as it currently stands. Therefore, we invite you to submit a revised version of the manuscript that addresses the points raised during the review process.

We look forward to receiving your revised manuscript.

Kind regards,

Md Jamal Uddin, Ph.D

Academic Editor

PLOS ONE

Journal Requirements:

4. We note that [Figures 2 and 4] in your submission contain [map/satellite] images which may be copyrighted. All PLOS content is published under the Creative Commons Attribution License (CC BY 4.0), which means that the manuscript, images, and Supporting Information files will be freely available online, and any third party is permitted to access, download, copy, distribute, and use these materials in any way, even commercially, with proper attribution. For these reasons, we cannot publish previously copyrighted maps or satellite images created using proprietary data, such as Google software (Google Maps, Street View, and Earth). For more information, see our copyright guidelines: http://journals.plos.org/plosone/s/licenses-and-copyright.

a. You may seek permission from the original copyright holder of Figures 2 and 4 to publish the content specifically under the CC BY 4.0 license.  

Reviewers' comments:

Reviewer's Responses to Questions

**Comments to the Author**

1. Is the manuscript technically sound, and do the data support the conclusions?

Reviewer #1: Yes

Reviewer #2: Yes

2. Has the statistical analysis been performed appropriately and rigorously? 

Reviewer #1: Yes

Reviewer #2: Yes

3. Have the authors made all data underlying the findings in their manuscript fully available?

Reviewer #1: Yes

Reviewer #2: Yes

4. Is the manuscript presented in an intelligible fashion and written in standard English?

Reviewer #1: Yes

Reviewer #2: No

5. Review Comments to the Author

Reviewer #1: Overall, the study presents promising insights, and the authors have made commendable efforts. However, before final publication, there is room for improvement. While the study's concept is strong, there is potential to refine its scope. A more concise version of the manuscript could enhance its impact. Given the extensive analyses conducted, emphasizing descriptive aspects might enhance clarity and readability.

Introduction

Within the global scenario in the first paragraph , it's essential to highlight specific statistics or trends that illustrate the magnitude of this issue.

Methodology

The use of Principal Component Analysis (PCA) to identify dietary patterns is a valid approach for analyzing dietary data. However, it would be beneficial to elaborate on how the specific food groups were chosen and categorized for PCA. Providing a clear rationale for grouping certain foods together can enhance the transparency of the analysis.

It would be helpful to know more about the specific statistical techniques used in the analysis, such as how the logistic regression model was constructed, including the inclusion and exclusion criteria for variables. Additionally, the methodology section could benefit from more information about how potential confounding factors were addressed, especially given the limitations in data availability.

Additionally, it's important to provide more information about the sample size and any potential biases in the data collection process. Mentioning how missing data were handled, if applicable, would also add clarity to the methodology.

Results

The manuscript revealed an inconsistency in the use of decimal points within the article. To ensure accuracy and reliable interpretation, efforts were made to rectify this inconsistency. Maintaining consistent and accurate decimal points throughout the study was prioritized to prevent any potential misinterpretations. This attention to detail aimed to uphold the integrity of the findings and maintain transparency in the reporting of results.

The submitted figures have a low resolution, making them difficult to interpret. Please consider providing higher resolution versions in the future for improved visibility and clarity.

Discussion

"This is the first study examining the association between dietary patterns and heart disease and their spatial heterogeneity using country-representative household income and expenditure datasets". It would be beneficial to omit the repeated sentence for the sake of clarity and flow of the content.

The discussion section effectively summarizes the findings and relates them to existing literature. To enhance the discussion, consider providing more context on why certain dietary patterns might be associated with heart disease risk. This could involve discussing the specific nutrients, bioactive compounds, or potential mechanisms underlying the observed associations.

Furthermore, mentioning the practical implications of the study's findings could add depth to the discussion. For example, discussing how these findings could inform public health policies, dietary guidelines, or interventions aimed at reducing heart disease risk in Bangladesh.

Grammar and Style

The text is generally well-written, but there are some instances of grammatical errors and typos. A thorough proofreading session can help catch and correct these issues, ensuring the text is clear and professional.

Reviewer #2: Thank you for the opportunity to review this important manuscript. This is an interesting topic. Authors have examined an important question.As a reviewer, I have thoroughly examined the manuscript and have provided feedback to enhance the overall quality and clarity of the research. In my evaluation, I have identified one major comment and four minor comments, which are outlined below:

Major comment:

1. In table 1, some of the variables (i.e., Age, Religion, Division) categories do not sum up to 100%.

Minor comments:

1. There are lots of grammatical and spelling mistakes throughout the paper. Try to recheck this paper and correct the mistakes.

For example, in line number 87 the word “aroud” should be “around”. In line number 102-103 data “would helpful” should be “would help or would be helpful”.

In line 55, “The dietary components significantly explain the prognosis of heart disease”

Error: "explain" should be replaced with "influence."

In line 60-62 "Conversely, a suboptimal diet enriched with trans-fats, sugar-sweeteners, refined grains, red meat, salt, and processed foods considered a core attributable factor for all causes of mortality and disability."

Error: Missing "is" before "considered."

2. “However, the majority of studies were carried out in western countries, making it challenging to extrapolate it for the dietary patterns of people residing in less developed countries due to disparate dietary habits, cultures, and availability (11).”

In line 77-79, mention some of the references that supports your sentence as you said that majority of the studies were carried out but you referenced only one.

3. In introduction section, the 2nd paragraph, you may add the worldwide scenario of heart related diseases and mortality with the variation of food habit.

4. There were no information about the types of heart-related diseases in Bangladesh and their prevalence in the introduction section.

6. PLOS authors have the option to publish the peer review history of their article (what does this mean?). If published, this will include your full peer review and any attached files.

Reviewer #1: No

Reviewer #2: No

---

## [Author Response · Author response to Decision Letter 0]

9 Nov 2023

Answers to the comments from the editor and reviewers:

Authors Response: The manuscript is formatted as per the journal guidelines.

Authors Response: In the revised cover letter data availability statement explain in detail. The statement is given below:

“This research was carried out using the 2016-2017 Bangladesh Household Income and Expenditure Survey (HIES) data which was conducted by the Bangladesh Bureau of Statistics (BBS) with technical and financial support from the World Bank. However, the BBS has imposed legal restrictions to prevent sharing the data publicly. Data is available upon request to the corresponding author with the permission from the BBS (Director General, Bangladesh Bureau of Statistics, dg@bbs.gov.bd, +88-02-5500-7056, www.bbs.gov.bd).”

Authors Response: Thank you for your valuable insight. In the revised version, we included the ethics statement in the “Methods” section- 

“Since the de-identified data for this study came from secondary sources, this study did not require any ethical approval. However, ethical approval was obtained for the HIES, and details can be found elsewhere [37]”

4. We note that [Figures 2 and 4] in your submission contain [map/satellite] images which may be copyrighted. All PLOS content is published under the Creative Commons Attribution License (CC BY 4.0), which means that the manuscript, images, and Supporting Information files will be freely available online, and any third party is permitted to access, download, copy, distribute, and use these materials in any way, even commercially, with proper attribution. For these reasons, we cannot publish previously copyrighted maps or satellite images created using proprietary data, such as Google software (Google Maps, Street View, and Earth). For more information, see our copyright guidelines: http://journals.plos.org/plosone/s/licenses-and-copyright.

Authors Response: To prepare Figure 2 and 4, we used a shape file which is publicly available at https://data.humdata.org/dataset/cod-ab-bgd. The source of this shape file is Bangladesh Bureau of Statistics (BBS) which is published under the Creative Commons Attribution for Intergovernmental Organizations (CC BY-IGO). Under the CC BY-IGO license, anyone can use (copy and redistribute the material in any medium or format) and/or adapt (remix, transform, and build upon the material) for any purpose, even commercially. The licensor cannot revoke these freedoms as long as one follows the license terms. The license terms are that one must give appropriate credit, provide a link to the license, and indicate if changes were made. Additionally, anyone may not apply legal terms or technological measures that legally restrict others from doing anything the license permits. When the Licensor is an intergovernmental organization, disputes will be resolved by mediation and arbitration unless otherwise agreed.

All of the requirements of CC BY-IGO are aligned with the Commons Attribution License (CC BY 4.0) requirement. That is why we do not need to take any permission and we can freely use this shape file to make our map.

Answers to the comments from the reviewer #1

Comment #1: Overall, the study presents promising insights, and the authors have made commendable efforts. However, before final publication, there is room for improvement. While the study's concept is strong, there is potential to refine its scope. A more concise version of the manuscript could enhance its impact. Given the extensive analyses conducted, emphasizing descriptive aspects might enhance clarity and readability.

Response: Thank you so much for your valuable insight and appreciation.

Introduction

Comment #2: Within the global scenario in the first paragraph, it's essential to highlight specific statistics or trends that illustrate the magnitude of this issue.

Authors Response: Thanks for your suggestion. We have updated the first paragraph with providing statistics and trends of heart disease. Our updated first paragraph are given below:

“Heart disease is escalating globally, a number one cause of death and disabilities that exacerbate health and well-being [1]. Over the past decades, the rate of heart disease has increased by 103%, affecting 197 million people and contributing to 182 million disability-adjusted life years and 9.1 million deaths (16% of total deaths) in 2019 [1]. The low and middle-income countries (LMICs), bear the greatest burden from the boom in heart disease rate due to poor medical care and other lifestyle-related factors, with over 80% of all heart disease-related deaths and disabilities [2]. South Asian countries have the highest burden and mortality from heart disease among all LMICs (40–60% higher risk of mortality), accounting for more than one-fourth of cardiovascular diseases (CVDs)-related deaths [3, 4]. Furthermore, South Asian ethnicities are an independent risk factor of heart disease, with a 3 to 5 times higher chance of developing heart disease, and heart disease manifests 5 to 10 years earlier than in Western countries. However, the risk was highest among Bangladeshis and exposed to heart disease at an earlier age [5]. Depending on context, the prevalence of heart disease rates in Bangladesh varies significantly, ranging from 1.85 to 78% [6, 7]. According to a systematic review and meta-analysis, the weighted pool prevalence of coronary heart disease in Bangladesh was 4%, ischemic heart disease was 2%, and heart attack was 2% [8].”

Methodology

Comment #3: The use of Principal Component Analysis (PCA) to identify dietary patterns is a valid approach for analyzing dietary data. However, it would be beneficial to elaborate on how the specific food groups were chosen and categorized for PCA. Providing a clear rationale for grouping certain foods together can enhance the transparency of the analysis.

Authors Response: Thank you for spotting the concern. In our revised version, we provided the rationale of food grouping. 

The HIES 2016 questionnaire had a distinct section with 133 lines of food items under 17 food groups designed to collect food consumption data over the preceding 14 days through 7 enumerators' visits. Along with 133 distinct food items, each food group had an entry labelled “Other”. This information allowed us to calculate each food type's daily intake (in grams). A total of 123 food items and “Other” were considered in this study, excluding “tobacco and tobacco products” and “betel leaf and chew goods” items. All food items were grouped into 27 groups based on similarities in nutritional components, and the food groups used in earlier studies in Bangladesh [27, 37]. Due to the difficulties in determining particular food items from each of the "Other" categories designated in each food group in HIES data base, these ‘Other’ were termed as "Other grains" and "Other vegetables" and were included in the analysis. Details of food groupings were also provided in the supplementary table which could enhance the transparency of the analysis (S1 Table).

Comment #4: It would be helpful to know more about the specific statistical techniques used in the analysis, such as how the logistic regression model was constructed, including the inclusion and exclusion criteria for variables. Additionally, the methodology section could benefit from more information about how potential confounding factors were addressed, especially given the limitations in data availability.

Authors Response: Thanks for the suggestion. We have provided more information in the statistical analysis section. 

In this study, the inclusion criteria all possible covariates for the multivariate logistic model were having p< 0.25 in the bivariate model [42]. In addition, the model fitness was checked by the Hosmer-Lemeshow goodness of fit test (p=0.88). Variance Inflation Factors (VIF) checked multicollinearity before conducting multivariate logistic regression and found VIF<3 indicating no collinearity issues [43].

Comment #5: Additionally, it's important to provide more information about the sample size and any potential biases in the data collection process. Mentioning how missing data were handled, if applicable, would also add clarity to the methodology.

Authors Response: In our revised version, we provided more information about sample size including key indicator on which sample size was calculated, sampling strategy. We also provided information on how data quality was ensured. We detailed up the missing data handling procedure. Participants were excluded from this study if missing data on any of the major variables—age, disease conditions, dietary information, and energy outlier—was found. However, there was a few missing data regarding the education and employment status of participants which was kept missing during further analysis.

Results

Comment #6: The manuscript revealed an inconsistency in the use of decimal points within the article. To ensure accuracy and reliable interpretation, efforts were made to rectify this inconsistency. Maintaining consistent and accurate decimal points throughout the study was prioritized to prevent any potential misinterpretations. This attention to detail aimed to uphold the integrity of the findings and maintain transparency in the reporting of results.

Authors Response: We apologize for this inconsistency. We have addressed all inconsistencies regarding decimal points in this revised version.

Comment #7: The submitted figures have a low resolution, making them difficult to interpret. Please consider providing higher resolution versions in the future for improved visibility and clarity.

Authors Response: Thank you for spotting the problem. We have added all figures with better resolution in the revised version. Journal has a specific criterion for maximum resolution. All figures are constructed with the highest resolution to meet the journal requirement.

Discussion

Comment #8: "This is the first study examining the association between dietary patterns and heart disease and their spatial heterogeneity using country-representative household income and expenditure datasets". It would be beneficial to omit the repeated sentence for the sake of clarity and flow of the content.

Authors Response: This line has been written in the following way to avoid repetition:

Dietary factors can play a significant role in either promoting or aggravating the pathophysiology of heart disease. This study explored the dietary patterns and risk of heart disease in Bangladesh. Using principal component analysis this study identified three major dietary patterns: “festival pattern”, “pickles and fast foods pattern”, and “rice and vegetable pattern”.

Comment #9 The discussion section effectively summarizes the findings and relates them to existing literature. To enhance the discussion, consider providing more context on why certain dietary patterns might be associated with heart disease risk. This could involve discussing the specific nutrients, bioactive compounds, or potential mechanisms underlying the observed associations.

Authors Response: In the revised version, we have provided a short mechanism of why certain dietary pattern was associated with heart disease. The mechanism is given in the fourth paragraph of discussion:

 “Fast food, deep-fried and sugary snacks usually contain a higher amount of refined carbohydrates, sugars, fats, saturated fats, cholesterols, and lower micronutrients, all of these ingredients are known to be atherogenic [51, 52]. These foods are usually prepared in an unhealthy way such as deep frying in oil, reusing oil, or cooking for an extended period, resulting in increased trans-fat which negatively affects lipid metabolism, increased inflammatory and oxidative stress, impaired endoplasmic function, apoptosis, and autophagy [53]. Besides, lower micronutrients, particularly vitamin B are linked to hyperhomocysteinemia, a well-known risk factor for coronary heart disease [6]. In addition to the high sugar content, refined grains with a high glycemic load can elevate inflammatory markers resulting in the development atherosclerosis. These high glycemic foods are also responsible for type 2 diabetes and obesity which are the established risk factors of heart disease [54]. Conversely, vegetables and spices contain numerous bioactive substances (vitamins, minerals, fibers, phytonutrients) which have been shown to have a cardioprotective role through a variety of mechanisms including regulating blood pressure, insulin sensitivity, and lipid metabolism; anti-oxidation, anti-inflammatory, and anti-platelet properties; better endothelial function, reducing myocardial damage, modulation of enzymatic activities, gene expressions and other signaling pathways [55, 56]. Furthermore, those who consume more fruits and vegetables usually lead a healthy lifestyle with more physical activity and less smoking, which may result in a reduced likelihood of heart disease [57]. Adequate fruits and vegetables consumption could lower the global burden of ischemic heart disease by 31% [58]. Hence, the dietary guidelines should prioritize the adoption of healthy eating practices such as Mediterranean dietary patterns, Prudent patterns, DASH diet, and plant-based dietary patterns. These emphasize the consumption of essential healthy food items such as fruits and vegetables, whole grains, legumes, fish, poultry, while discouraging unhealthy eating practices seen in Western dietary patterns characterized by higher intake of refined grains, red meat, fast foods, processed foods, and sugar [10].”

Comment #10. Furthermore, mentioning the practical implications of the study's findings could add depth to the discussion. For example, discussing how these findings could inform public health policies, dietary guidelines, or interventions aimed at reducing heart disease risk in Bangladesh.

Authors Response: The implication of the study is also provided in the revised version which has been attached in the following lines:

 “This study identified the prevailing dietary patterns across the country, providing valuable insights for the government to develop effective guidelines for tailoring hear

---

## [Decision Letter · Decision Letter 1]

5 Jun 2024

PONE-D-23-21261R1Geospatial variation in dietary patterns and their association with heart disease in Bangladeshi population: Evidence from a nation-wide surveyPLOS ONE

Dear Dr. Amin,

Thank you for submitting your manuscript to PLOS ONE. After careful consideration, we feel that it has merit but does not fully meet PLOS ONE’s publication criteria as it currently stands. Therefore, we invite you to submit a revised version of the manuscript that addresses the points raised during the review process.

We look forward to receiving your revised manuscript.

Kind regards,

Mohammad Nayeem Hasan

Academic Editor

PLOS ONE

Reviewers' comments:

Reviewer's Responses to Questions

**Comments to the Author**

1. If the authors have adequately addressed your comments raised in a previous round of review and you feel that this manuscript is now acceptable for publication, you may indicate that here to bypass the “Comments to the Author” section, enter your conflict of interest statement in the “Confidential to Editor” section, and submit your "Accept" recommendation.

Reviewer #3: All comments have been addressed

Reviewer #4: (No Response)

2. Is the manuscript technically sound, and do the data support the conclusions?

Reviewer #3: Yes

Reviewer #4: Yes

3. Has the statistical analysis been performed appropriately and rigorously? 

Reviewer #3: Yes

Reviewer #4: Yes

4. Have the authors made all data underlying the findings in their manuscript fully available?

Reviewer #3: Yes

Reviewer #4: No

5. Is the manuscript presented in an intelligible fashion and written in standard English?

Reviewer #3: Yes

Reviewer #4: Yes

6. Review Comments to the Author

Reviewer #3: To my knowledge, the authors have done a good job preparting this manuscript and I've found the quality and the technicality of this research to the mark. They have mentioned all the comments from the previous round of review and this manuscript can now be accepted.

Reviewer #4: Thanks for the beautiful work for Bangladesh people,

Please work on the following points:

1- How do you proof you received the data, what is the document states that you got the data?

2- It is not so good to apply a survey result which is done in 2016 for your people 2024, how curiously they read the article and accept the results?!. Where not any new data? If not, why did not you collected a newer version? You could choose some criteria that is suitable and allowed from the your local government.

3- Line 127 Variables of the study, the section of education is too wide since higher levels of education result in better knowledge of keeping self against health problems.

4- This is not nice to put link in the middle of an article like in line 191 online(https://data.humdata.org/dataset/cod-ab-bgd) using Arc GIS 10.8.1 where geospatial... Please make a reference to it.

5- Lines 216-224 does not need bold and link, smoothly talk about figure 2.

6- While the data collected in Bangladesh, it does not need to mention all the time in Bangladesh like in the table heads and section heads.

7- Please move line 267 Table 4. Characteristics of the study participants by tertile categories of dietary patterns. to the table page.

8- Once again, lines 277 and 286-292 is bold and only need to mention the figures easily.

9- The conclusion section is too short.

10- I would like you to suggest some solutions and methods to your people in Bangladesh to protect themselves from heart diseases and add the suggestions in the discussion or conclusion part.

11- Please add the DOI link or website link for reference number 16, 25, 26, 32, 37, 38, 39, 43, and 46 to maintain the uniformity of the references.

12-Most of the references look like they have been brought from local authors, please add some major references especially the new ones, after 2022.

7. PLOS authors have the option to publish the peer review history of their article (what does this mean?). If published, this will include your full peer review and any attached files.

Reviewer #3: No

Reviewer #4: **Yes: **Askandar Hamid Amin

---

## [Author Response · Author response to Decision Letter 1]

4 Jul 2024

We appreciate the constructive feedback provided by the respected reviewers on our manuscript titled “Geospatial variation in dietary patterns and their association with heart disease in Bangladeshi population: Evidence from a nation-wide survey.” We have carefully considered each comment and have made revisions accordingly. Please find below our detailed responses to each point raised:

Reviewer #4: Thanks for the beautiful work for Bangladesh people, Please work on the following points:

1. Comment: How do you proof you received the data, what is the document states that you got the data?

Response: Thank you for your concern. We purchased the datasets from the Bangladesh Bureau of Statistics (BBS). Md Ruhul Amin and Md Masum Ali, two of our co-authors, were engaged in this process. To verify our data acquisition, we have attached the payment receipt.

2. Comment: It is not so good to apply a survey result which is done in 2016 for your people 2024, how curiously they read the article and accept the results?!. Where not any new data? If not, why did not you collected a newer version? You could choose some criteria that is suitable and allowed from the your local government.

Response: Thank you for your insightful comments. We understand the concern regarding the use of a 2016 dataset for a study conducted in 2024. Unfortunately, there are no more recent datasets available that meet the necessary criteria for this research. Despite the dataset being from 2016, it remains highly relevant. It provides valuable insights into the topic under investigation, and the change of dietary patterns at the population level takes a longer period. The authors thought that it was okay to answer research questions posed in this paper using this data set. Further, we ensured rigorous analysis and interpretation to draw meaningful conclusions that can still be applicable today. Additionally, the 2016 dataset is comprehensive, and national representative, making it a suitable foundation for our study. We hope this addresses your concerns and highlights the significance of our findings.

3. Comment: Line 127 Variables of the study, the section of education is too wide since higher levels of education result in better knowledge of keeping self against health problems.

Response: Yes, I agree with you. Therefore, education was adjusted to see the effects of dietary patterns on heart disease. Further, the literature supports the inclusion and categorization of education variables for this study. The category used in this paper is widely used in Bangladesh, as can be seen in the following references of our manuscript (29, 32):

1. Sarwar N, Ahmed T, Hossain A, Haque MM, Saha I, Sharmin KN. Association of dietary patterns with type 2 diabetes mellitus among Bangladeshi adults. International Journal of Nutrition Sciences. 2020 Dec 1;5(4):174-83. doi: 10.30476/IJNS.2020.88173.1091

2. Khanam F, Hossain MB, Mistry SK, Afsana K, Rahman M. Prevalence and Risk Factors of Cardiovascular Diseases among Bangladeshi Adults: Findings from a Cross-sectional Study. J Epidemiol Glob Health. 2019 Sep;9(3):176-84. doi: 10.2991/jegh.k.190531.001 

4. Comment: This is not nice to put link in the middle of an article like in line 191 online(https://data.humdata.org/dataset/cod-ab-bgd) using Arc GIS 10.8.1 where geospatial... Please make a reference to it.

Response: Thank you for your feedback. We removed the link from the middle of the article and added it to the references section instead, which can be found in line 217. The added reference is given below:

OCHA. Bangladesh - Subnational Administrative Boundaries; 2020 [cited 2024 Jun 12]. Available from: https://data.humdata.org/dataset/cod-ab-bgd

5. Comment: Lines 216-224 does not need bold and link, smoothly talk about figure 2.

Response: Thank you for your nice feedback. In the revised manuscript, we removed the link, which can be found on lines 244-248. Furthermore, the bolded line is the title of the figure. According to the PLOS ONE submission guidelines, “each figure caption should appear directly after the paragraph in which it is first cited, with the figure title in bold. The figure legend should follow the title in normal text”. Therefore, to adhere to these guidelines, we bolded the figure titles.

6. Comment: While the data collected in Bangladesh, it does not need to mention all the time in Bangladesh like in the table heads and section heads.

Response: Thank you for your nice comment. We agree with you. Now, in the revised manuscript, we removed “in Bangladesh” from tables and section heads to prevent the overuse of the phrase. 

7. Comment: Please move line 267 Table 4. Characteristics of the study participants by tertile categories of dietary patterns. to the table page.

Response: Thank you. This has been moved to the second revised manuscript as can be seen in line 288.

8. Comment: Once again, lines 277 and 286-292 is bold and only need to mention the figures easily.

Response: Thank you for your suggestion. The bolded lines (298, 306) are the title of the figures. According to the PLOS ONE submission guidelines, “each figure caption should appear directly after the paragraph in which it is first cited, with the figure title in bold. The figure legend should follow the title in normal text.” Therefore, to adhere to these guidelines, we bolded the figure titles.

9. Comment: The conclusion section is too short.

Response: Thank you for your insightful comment. The conclusion has been rewritten in the revised version.

10. Comment: I would like you to suggest some solutions and methods to your people in Bangladesh to protect themselves from heart diseases and add suggestions in the discussion or conclusion part.

Response: Thank you for your suggestions. Here, we think that you might examine the first submission of the manuscript. However, during the first revision, we addressed this issue. We added some suggestions/solutions and policy implications for Bangladeshi people, which can be found on lines 423-431:

“This study identified the prevailing dietary patterns across the country, providing valuable insights for the government to develop effective guidelines for tailoring heart disease prevention programs. Our study recommends a public awareness campaign to promote a heart-healthy dietary choice, such as incorporating rice, vegetables, fruits, and spices, while discouraging the consumption of fast food and sugary items through mass media, educational institutions, and other channels. Furthermore, the government should impose a tax on unhealthy food items and provide subsidies on heart-healthy food items to ensure affordability for the general population. As this study identified the vulnerable regions to heart disease and unhealthy dietary patterns, it can provide valuable insights for policymakers for tailoring region-specific interventions.”

11. Comment: Please add the DOI link or website link for reference number 16, 25, 26, 32, 37, 38, 39, 43, and 46 to maintain the uniformity of the references.

Response: Thank you for your suggestion. In the revised version, we have followed the guidelines of PLOS ONE for references. We included the URLs of the references where appropriate and possible, according to the PLOS ONE guidelines. Some of the references you spotted are books or book sections, which have been formatted according to the PLOS ONE guidelines, and these guidelines do not require a DOI or website link for such references. Furthermore, for published articles without DOIs, the PLOS ONE guidelines indicate that they should be presented without a DOI or link.

12. Comment: Most of the references look like they have been brought from local authors, please add some major references especially the new ones, after 2022.

Response: Thank you for your constructive feedback. In the introduction section, we provided the available evidence on related topics in Bangladesh to identify the research gap. In the discussion, we compared the findings of our study with earlier studies conducted in Bangladesh, which resulted in the inclusion of several studies from local authors. However, following your suggestion, we have carefully reviewed the literature and added some recent related references from the literature. The following references are added (1, 15, 22, 48, 50, 53):

1. Martin SS, Aday AW, Almarzooq ZI, Anderson CA, Arora P, Avery CL, et al. 2024 Heart Disease and Stroke Statistics: A Report of US and Global Data From the American Heart Association. Circulation. 2024 Feb 20;149(8):e347-e913. doi: 10.1161/CIR.0000000000001209.

2. Wei L, Fan J, Dong R, Zhang M, Jiang Y, Zhao Q, et al. The Effect of Dietary Pattern on Metabolic Syndrome in a Suburban Population in Shanghai, China. Nutrients. 2023 May 4;15(9):2185. doi: 10.3390/nu15092185

3. Proma AY, Das PR, Akter S, Dewan SMR, Islam MS. The urgent need for a policy on epidemiological data on cardiovascular diseases in Bangladesh. Health Sci Rep. 2023 Jul 7;6(7):e1410. doi: 10.1002/hsr2.1410

4. Ahmed W, Muhammad T, Maurya C, Akhtar SN. Prevalence and factors associated with undiagnosed and uncontrolled heart disease: A study based on self-reported chronic heart disease and symptom-based angina pectoris among middle-aged and older Indian adults. PLoS One. 2023 Jun 28;18(6):e0287455. doi: 10.1371/journal.pone.0287455

5. Ram S, Chandra R, Kundu A, Singh A, Singh S, Tanti A, Bhattacharjee B, Tripathi P. Prevalence and determinants of self-reported heart disease among Indian men aged 15–54 years: Evidence from NFHS-5. Clinical Epidemiology and Global Health. 2023 Sep 1;23:101374.

6. Islam S, Nowar A, Amin MR, Shaheen N. Cost of Recommended Diet (CoRD) and Its Affordability in Bangladesh. Foods. 2023 Feb 13;12(4):790. doi: 10.3390/foods12040790

Phone: +8801819504434

ruhul.infs@du.ac.bd

---

## [Editor Report · Decision Letter 2]

8 Jul 2024

Geospatial variation in dietary patterns and their association with heart disease in Bangladeshi population: Evidence from a nationwide survey

PONE-D-23-21261R2

Dear Dr. Md. Ruhul Amin,

We’re pleased to inform you that your manuscript has been judged scientifically suitable for publication and will be formally accepted for publication once it meets all outstanding technical requirements.

Kind regards,

Mohammad Nayeem Hasan

Academic Editor

PLOS ONE
---

## [Editor Report · Acceptance letter]

10 Jul 2024

PONE-D-23-21261R2 

PLOS ONE

Dear Dr. Amin, 

I'm pleased to inform you that your manuscript has been deemed suitable for publication in PLOS ONE. Congratulations! Your manuscript is now being handed over to our production team.

Kind regards, 

on behalf of

Dr. Mohammad Nayeem Hasan 

Academic Editor

PLOS ONE